# Construction of Personalized Bus Travel Time Prediction Intervals Based on Hierarchical Clustering and the Bootstrap Method

Zhenzhong Yin [1],* and Bin Zhang [2],*

[1] School of Computer Science and Engineering, Northeastern University, Shenyang 110819, China
[2] Software College, Northeastern University, Shenyang 110819, China
* Correspondence: yinzhenzhong@stumail.neu.edu.cn (Z.Y.); zhangbin@mail.neu.edu.cn (B.Z.)

**Abstract:** Providing accurate bus travel time information is very important to help passengers plan their trips and reduce waiting times. Due to the uncertainty of the bus travel time, the traditional prediction value of the travel time point cannot accurately describe the reliability of the prediction result, which is not conducive to passengers waiting for the bus according to the prediction result. At the same time, due to the large differences in the individual driving styles of the bus drivers, the travel time data fluctuate greatly, and the accuracy and reliability of the point prediction results are further reduced. To address this issue, this study develops a personalized bus travel time prediction intervals model for different drivers based on the bootstrap method. Personalized travel time prediction intervals were constructed for drivers with different driving styles. To further improve the quality of travel time prediction intervals, this study optimizes training data sets considering driving style factors. Then, this paper integrates hierarchical clustering, an artificial neural network, and the bootstrap method to construct another prediction intervals model for bus travel time based on driver driving style clustering and the bootstrap method. The real−world driving data sets of the No. 239 bus in Shenyang, China, were used for experimental verification. The results showed that the two models constructed in this paper can effectively quantify the uncertainty of the point prediction results, the *PICP* of each interval exceeding the confidence level set (80%). It was also found that the quality of the prediction intervals constructed by clustering the driving style data is better (*MPIW* values decreased by 23.33%, 54.24%, and 28.61 respectively, and the corresponding *NMPIW* values also decreased by 18.93%, 10.39%, and 14.19%, respectively), which can provide passengers with more reasonable suggestions for waiting time.

**Keywords:** bus travel time prediction; prediction intervals; bootstrap; hierarchical clustering; driving style





## 1. Introduction

In an intelligent transportation system, accurate bus travel time information is very important for passengers to arrange their waiting time and reasonably select routes [1,2]. However, due to the complexity of urban traffic, prediction of bus travel time has become a very challenging topic [3]. The travel time of buses is affected by many factors, such as changes in passenger flow [4], traffic conditions [5,6], space–time factors [7–9], signals [10–12], driver driving styles [13], and weather factors [5,14]. Especially in some developing countries that lack lane discipline, all types of vehicle (such as private cars, buses, trucks, tricycles, and bicycles) share the road, and there is no isolation between them, which leads to a high degree of uncertainty and variability in bus travel time. This has caused negative impacts, such as bus bunching, longer waiting times for passengers, and increased bus operating costs [15,16].

In previous studies, a variety of bus travel time prediction methods, such as multiple linear regression, artificial neural network [17,18], support vector regression [18,19], Kalman

filter [17], random forest [13], and other machine learning models, have been proposed. However, most of the research mentioned above is focused on improving point prediction models. The main disadvantage of the traditional point prediction model is that it cannot reflect the uncertainty of the prediction result and cannot provide information, such as the confidence level or reliability. Even current state−of−the−art models can only minimize the error and cannot perfectly and accurately predict the travel time of buses [20]. The point−predicted result of bus travel time may be greater or less than the true value. When the predicted time is less than the true value, the passenger has to wait for the bus. Passengers may miss the bus when the predicted time is greater than the true value.

The uncertainty of the prediction of bus travel times can be described and measured by prediction intervals (PIs). PIs are a statistical tool that quantify the uncertainty associated with prediction by estimating the travel time range. Therefore, from the perspective of passengers, it is more meaningful to predict the possible range of bus arrival times than to predict only a single time point [3]. Figure 1 shows an example to explain the comparison of the effect of providing point prediction and PIs at a bus station. In this example, suppose that the passengers are ready to wait at station B, and the next bus has arrived at station A at this time (8:00). According to the point prediction results (blue dotted line in the middle), it takes about 7 min for the bus to arrive at station B from station A, that is, it is about 8:07 when it arrives at station B. Due to the uncertainty of the point prediction results, passengers may miss the bus when they arrive at the station on time at 8:07. However, this will not happen according to the PI results (green and red dotted lines). If passengers can know the PIs of the next bus arrival time at station B in advance, such as 8:05~8:10, passengers can plan to arrive at station B at 8:05 to wait, which can ensure that they will not miss the bus. As can be seen from the comparative examples, the travel time PIs can provide passengers with more reliable waiting time suggestions than a single−point prediction value.

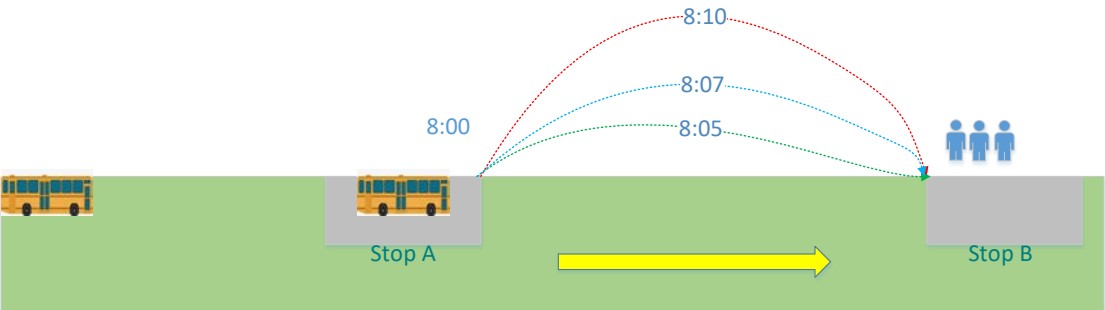

**Figure 1.** An example of PIs and point prediction for bus travel time.

However, only a few studies have dealt with the issue of predicting the variability of bus travel time. For example, in [21], the Bayesian method and the Delta method were used to construct PIs for bus and freeway travel times. The results show that the Delta method is better than the Bayesian method regarding the strictness of PIs. The PIs constructed by Bayesian technology have stronger robustness and better coverage probability than the neural network structure. In [3], the author developed a method based on genetic algorithms, which can automatically select neural network models and adjust super parameters. The bus and freeway travel time data sets were used in experiments and results show that the proposed method can improve the quality of the PIs' width and coverage probability. A bootstrap method based on maximum likelihood technology was proposed to construct PIs in [22], where they discussed the possible sources of uncertainty in neural network prediction, and proposed the concepts of confidence intervals and PIs and the techniques of quantifying them. Then the method proposed was applied to predict the bus travel time along a bus line in Melbourne, Australia, and its performance was evaluated.

However, previous researchers have not added driving style factors to the study of bus travel time PI prediction. To our knowledge, there is no research on personalized bus travel time PI models. Driving style factors can affect bus travel time, which has been mentioned in some studies [13,23]. The essential part of driving style analysis is collecting driving data. However, the driving data used in many studies were collected from questionnaires, site investigations, or laboratory simulations, which may differ from real driving environments and result in biased outcomes [24]. Drawing on the strengths of GPS, big data, and machine learning, this paper proposes a bus personalized travel time PI model based on the bootstrap method, which can predict the personalized travel time intervals for different drivers.

Due to the sparseness of the number of buses, the quality of PIs predicted by using a single driver's travel time data set is not ideal. Therefore, to further improve the quality of a travel time PI (minimize its width and improve its coverage probability), we optimize the training data set by considering driver driving style factors. Based on the above prediction model, this paper integrates hierarchical clustering, an artificial neural network, and the bootstrap method to construct another PI model of bus travel time. This model firstly clusters the travel time data of multiple different drivers according to the driving style, and secondly uses the bootstrap method to sample the clustered data. Finally, it predicts the travel time of the sampled data, and calculates the upper and lower limits of the travel time PIs.

The main contributions of this paper can be summarized as follows:

- We establish a personalized bus travel time PI model based on the bootstrap method for drivers with different driving styles. The proposed model can provide accurate and stable prediction results.
- To the best of our knowledge, this paper is the first to study the influence of driving style factors on the quality of bus travel time PIs.
- In order to improve the quality of PIs, we use the hierarchical clustering method to cluster the travel time data sets of multiple different drivers according to driving style, and further optimize the quality of personalized travel time PIs based on the bootstrap method.

The remainder of this paper is arranged as follows. A brief review of the related works is given in Section 2. Section 3 briefly introduces the process of the bootstrap method used to construct PIs. Section 4 outlines the methodological framework and presents the construction process of the PI models of the bus travel time. In Section 5, the effectiveness of the proposed method is verified by experiments. Finally, we provide conclusions in Section 6.

## 2. Related Works

As mentioned earlier, bus travel time PI information is crucial for traffic managers and passengers. To build personalized travel time PIs for bus drivers, it is necessary to classify drivers with different driving styles and then predict travel time intervals for different types of drivers. Therefore, this section discusses the relevant research in two parts: driving style classification and PI construction.

### 2.1. Review of Classification of Driver's Driving Style Research

The driver's driving style is understood as the way the driver operates the vehicle controls in the context of the driving scene and external conditions such as time of the day, day of the week, weather, and mood, among other factors [25]. Previous studies have shown that drivers have a range of different driving styles, from calm or defensive, to aggressive or assertive [26–30]. The driving style is manifested mainly in the control of vehicle speed, acceleration, lane changes, headway, and other aspects during driving [31]. The most common aggregate driving behavior parameters used in research are mean speed, mean positive speed, mean acceleration, mean deceleration, mean driving duration, and number of acceleration and deceleration events [28,31–33].

Driving style recognition algorithms are usually divided into three categories: (1) Threshold−based algorithms [33,34]. This kind of algorithm is simple to use, but the accuracy is quite limited. (2) Model−based algorithms [32,35]. The main disadvantage of these algorithms is that it is difficult to prove the accuracy of the results. (3) Machine learning algorithms include, but are not limited to, unsupervised machine learning algorithms such as the hierarchical clustering analysis algorithm [36], principal component analysis algorithm [36], Gaussian mixture model (GMM) [35], and other unsupervised machine learning algorithms; and K−nearest neighbor (KNN) [37,38], neural network (NN) [37,39], and other supervised machine learning algorithms. The use of machine learning algorithms to identify driving styles has become the mainstream method. In this work, a hierarchical clustering analysis model is proposed to classify the driving styles of bus drivers. Hierarchical clustering (HC) is an unsupervised learning method and a statistical method that is used to divide objects into groups with similar meanings. When researchers do not know the number of groups in advance and want to establish groups and analyze group members, hierarchical clustering analysis is usually used. It seeks to identify groups that both minimize intergroup variation and maximize outer−group variation [36].

### 2.2. Review of PI Forecasting Literature

The PIs consist of upper and lower limits, in which the actual target lies within these limits and has a specified probability called the confidence level. The confidence level is usually described as $((1 - a)\%)$, where $a$ changes from 0 to 1. Wider PIs indicate that the uncertainty associated with the prediction is higher, so decisions should be made more carefully based on the results. On the contrary, when the constructed PIs have a narrow width, decisions can be made with more confidence [40]. In the literature, techniques for construction of PIs mainly include: the Bayesian technique [21], which is based on Bayesian statistics and has a strong mathematical foundation, but requires the calculation of complex matrices; the Delta technique [3,21,40], which is based on the representation and interpretation of NNs as a nonlinear regression model; it allows the application of standard asymptotic theory to them to construct PIs, which also requires the calculation of complex matrices; the bootstrap method [22,40], which is a PI construction technique that is widely used in the literature, and is essentially a resampling method that can produce good PIs with high probability of coverage; the main drawback of this method is that it requires a large amount of computation during the development and utilization stage; the upper and lower bound estimation method (LUBE) [41], which is prone to fall into local minima and cannot obtain satisfactory results. The application of these techniques has been documented in the literature in the fields of transportation [21,26], manufacturing [22,41], load forecasting [21,22], renewable energies [40,41], and electricity prices [40]. A comprehensive review and a detailed discussion of these methods can be found in the review article [42].

In this study, the bus departure time is distributed across different times in each day. The distribution of departure times includes peak hours and flat peak hours, so the difference in the travel times of the same section caused by different departure times is also relatively large. Therefore, the travel time data of the same driver on the same road section in the same period of time (the period is about one month) represent a typical small sample.

For small numbers of data points (e.g., 100 or less), the PIs based on the bootstrap method tend to be more accurate [43], given that they do not rely on asymptotic results but on the construction of the limit distribution from the available data [44]. The bootstrap PI method can repeatedly resample the original data samples by computer, convert small samples into large samples to estimate the approximate distribution of unknown parameters, and then construct the PIs [3]. According to this, the main focus of this research is on the use of the bootstrap technique for PI construction.

### 3. Bootstrap Method for PI Construction

The bootstrap method is a data resampling technique based on computer technology proposed by Professor Efron in 1979 [45]. It can be used to study the distribution characteristics of a statistic in a group of data. It is a commonly used method for constructing confidence intervals and PIs. Compared to other PI construction technologies, the main advantage of bootstrap technology is its simplicity. Compared to Delta and Bayesian technology, this method does not need to calculate complex matrices [22].

Referring to the literature [22,46], suppose there is a set of data $D = \{(x_i, y_i), i = 1, 2 \ldots N\}$, in which there is a nonlinear mapping relationship $y(x_i)$ between the target value $y$ and the input variable $x$. Considering the error of travel time measurement, the travel time can be expressed as:

$$y_i^* = y(x_i) + \epsilon(x_i) \tag{1}$$

where $y(x_i)$ represents the true value of the travel time of the $i$th sampling, $y_i^*$ indicates the corresponding measured value, and $\epsilon(x_i)$ represents noise, which can generally be assumed to obey a normal distribution with a mean value of 0. In the process of constructing the PIs, the output of the prediction model is assumed to be $\hat{y}_i(x_i)$; then, the error of the model can be expressed as [46]:

$$y_i^* - \hat{y}_i = [y(x_i) - \hat{y}_i(x_i)] + \epsilon(x_i) \tag{2}$$

The prediction error in Equation (2) includes two parts: $y(x_i) - \hat{y}_i(x_i)$ represents the error between the real value and the predicted value of the model, and $\epsilon(x_i)$ represents data noise, which can generally be considered to be independent of each other. The prediction variance of the prediction model can be expressed as [22]:

$$\sigma_y^2(x_i) = \sigma_{\hat{y}}^2(x_i) + \sigma_\epsilon^2(x_i) \tag{3}$$

According to Formula (3), if $\sigma_{\hat{y}}^2(x_i)$ and $\sigma_\epsilon^2(x_i)$ are estimated, $\sigma_y^2(x_i)$ can be calculated, so the PIs can be obtained according to the variance estimation value. The bootstrap method is used to estimate $\sigma_{\hat{y}}^2(x_i)$ and $\sigma_\epsilon^2(x_i)$.

The essence of the bootstrap method is the resampling process, which simulates the population distribution by resampling the observed data to generate regenerated samples. Suppose there is a data sample $X = \{x_1, x_2, \ldots x_n\}$ with a sample capacity of $N$, and $B$ samples with a capacity of $M$ are extracted from the original sample $X$. $X^* = \{x_1^*, x_2^*, \ldots, x_m^*\}$, usually $M = N$, finally yielding $B$ bootstrap samples $X_1^*, X_2^*, \ldots, X_B^*$. Since there is no strong correlation between the size of the $B$ value and the width of the PIs, a $B$ value that is too large cannot significantly improve the quality of the PIs. $B$ can meet the requirements of most practical engineering applications in the range of 20~200 [43], and $b = 30$ is selected in this paper.

The mean of the travel time point prediction using the ANN models for the $B$ samples is [46]:

$$\hat{y}(x_i) = \frac{1}{B} \sum_{b=1}^{B} \hat{y}_b(x_i) \tag{4}$$

where $\hat{y}_b(x_i)$ is the prediction of the $i$th sample generated by the bth model. Then, the variance of the prediction results of $B$ models is used to estimate the model variance [46].

$$\sigma_{\hat{y}}^2(x_i) = \frac{1}{B - 1} \sum_{b=1}^{B} (\hat{y}_b(x_i) - \hat{y}(x_i))^2 \tag{5}$$

We also need to estimate the variance of the error $\sigma_\epsilon^2(x_i)$, estimated by establishing a new learning model, which can be obtained from Equation (3):

$$\sigma_\epsilon^2(x_i) = E\left\{(y_i^* - \hat{y}(x_i))^2\right\} - \sigma_{\hat{y}}^2(x_i) \tag{6}$$

The sum of squares of residuals is as follows [46]:

$$r^2(x_i) = max((y_i^* - \hat{y}(x_i))^2 - \sigma_{\hat{y}}^2(x_i), 0) \tag{7}$$

where $\hat{y}(x_i)$ and $\sigma_{\hat{y}}^2(x_i)$ can be calculated from Equations (4) and (5). The residuals are combined with the input variable set samples to build a new data set: $D_{r2} = \left\{ (x_i, r^2(x_i)) \right\}_{i=1}^{n}$, and $x_i$ is the input variable of the model. Through data set $D_{r2}$, a new neural network learning model is trained to estimate $\sigma_\epsilon^2(x_i)$; the goal is to ensure the observed sample in $D_{r2}$ has the greatest probability of occurrence, using the maximum likelihood estimation method as the cost function to train the model [46]:

$$C_{BS} = \frac{1}{2} \sum_{i=1}^{n} \left( ln\left( \sigma_\epsilon^2(x_i) \right) + \frac{r^2(x_i)}{\sigma_\epsilon^2(x_i)} \right) \tag{8}$$

After estimating $\sigma_{\hat{y}}^2(x_i)$ and $\sigma_\epsilon^2(x_i)$, the PIs with confidence level $(1 - \alpha)\%$ can be calculated as follows [22,46]:

$$\hat{y}(x_i) \pm t_{df}^{1-\frac{\alpha}{2}} \sqrt{\sigma_{\hat{y}}^2(x_i) + \sigma_\epsilon^2(x_i)} \tag{9}$$

where $t_{df}^{1-\frac{\alpha}{2}}$ is the $1 - \alpha/2$ quantile of the t distribution function with the degree of freedom $df$, and the value of the degree of freedom $df$ can generally be set to $B$, which is equivalent to the number of models trained by the model variance estimation.

## 4. Methodology

### 4.1. Framework

The framework of the personalized bus travel time PI model based on the bootstrap method proposed in this paper is shown in Figure 2. The specific steps to construct the bus travel time PIs are as follows:

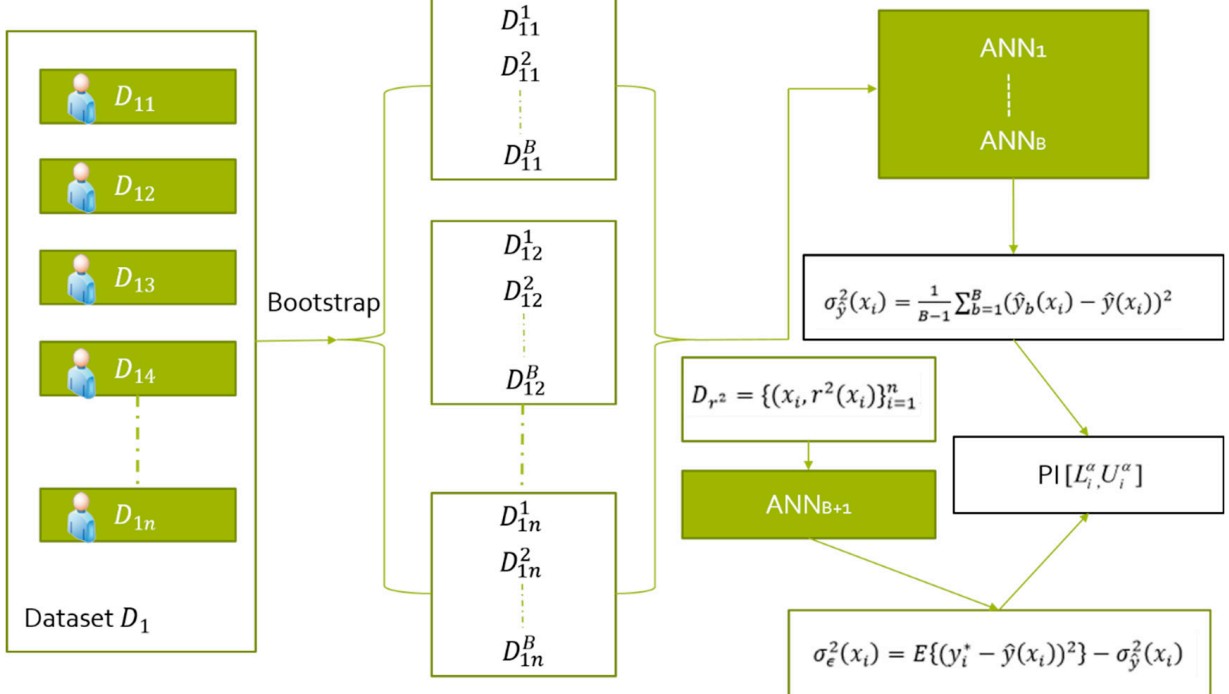

**Figure 2.** Model framework of bus personalized travel time PIs based on the bootstrap method.

(1) Build the original sample data set $D_1$. Data set $D_1$ contains n sub-sample sets $D_{11} \ldots D_{1n}$; each sub−sample set represents a driver's travel time history data set over a period of time.

(2) Bootstrap resampling. Each sub−sample set in the data set $D_1$ is resampled $B$ times ($B = 30$ in this paper) with replacement to obtain $B$ bootstrap samples.

(3) Construct a point prediction model. Each bootstrap sample is trained with ANN to obtain $B$ ANN models, and then the test samples are predicted to obtain $B$ prediction results and their variances.

(4) Construct a square residual data set. Construct the squared residual data set $D_{r^2} = \left\{ \left( x_i, r^2(x_i) \right) \right\}_{i=1}^{n}$ according to formula (7), and change the objective function of the ANN model to formula (8) to train on this data set and compute the variance of random errors on the test set.

(5) Construct PIs. Construct the corresponding PIs of travel time according to Equation (9).

We optimized the training data set by considering the driver's driving style factors to further improve the quality of PIs. A bus travel time PIs model based on driver driving style clustering and bootstrap was constructed by integrating hierarchical clustering technology on the basis of the above prediction model, as shown in Figure 3. First, the original data set $D_1$ obtains m new sub−sample sets ($D_{21} \ldots D_{2m}$) by hierarchical clustering according to driving style. Then, bootstrap sampling is carried out on the new data set after clustering, and finally PIs are determined according to the above steps.

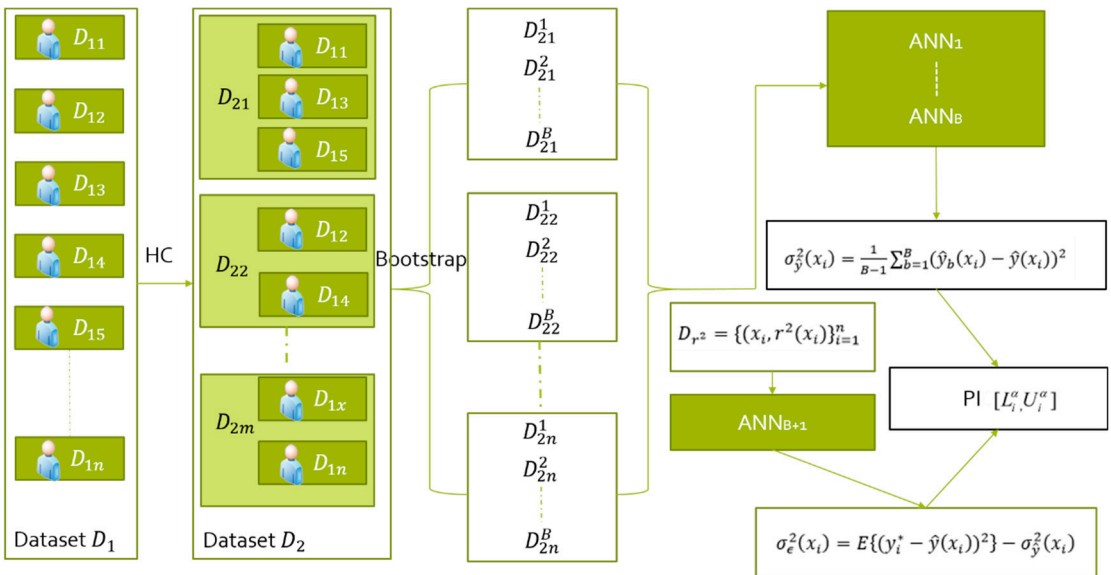

**Figure 3.** Model framework of bus travel time PIs based on driver driving style clustering and bootstrap.

### 4.2. Driver Driving Style Classification

To accurately classify the driving styles of bus drivers into different types, we consider both spatial and temporal features.

First, time period segmentation. Specifically, the day was separated into three time periods, that is, 7:00–9:00 (peak time), 9:00–16:00 (off−peak time), and 16:00–19:00 (peak time). The purpose of the time period separation is to ensure that the bus driver of travel times has similar patterns in the same time period. Because there are different traffic patterns in different time periods, different drivers have different behaviors in different traffic patterns, affecting travel time [19].

Second, segmentation of the bus route. In this step, the road segment between two adjacent bus stops is used as the basic prediction unit. The driving style of the driver can be classified by the travel time between two adjacent stops.

In our work, we intend to use a hierarchical cluster analysis model to classify the driving style of bus drivers. This method is formally described in Algorithm 1.

---

**Algorithm 1** Hierarchical Clustering Algorithm

---

Compute the proximity matrix.
Repeat
Merge the closest two clusters.
Update proximity matrix.
Until only one cluster remains.

---

### 4.3. Bus Travel Time Prediction

In this study, three layers of neural networks, i.e., the input layer, hidden layer, and output layer, are selected. The input layer is used to receive the training data set from the network. The output layer is a single neuron that outputs a travel time prediction.

It is very important to determine the appropriate traffic condition estimation factors to accurately predict bus travel time in a timely manner. The factors (inputs) considered in this study are described below.

(1) $X_1$: One day of the week. In general, the traffic flow is also different on the five working days of the week. For example, in Shenyang, traffic flow in the morning peak period on Monday and the evening peak period on Friday is significantly higher than that in other periods, and traffic intersections are more congested.

(2) $X_2$: Road segment number. The number of segments is referred to between two adjacent stations of the predicted bus line. Different sections of the road have different characteristics, for example, the length of the road section, traffic conditions, and the number of signal lights. All these differences may lead to changes in bus travel time.

(3) $X_3$: Departure time. It is defined as the predicted departure bus time. At different times of the day, the travel time of the bus is also different. For example, the travel time of buses varies greatly between peak hours and off−peak hours.

The bus travel time prediction model can be summarized in the following form:

$$T_{predicted} = f(X_1, X_2, X_3) \tag{10}$$

## 5. Experiments and Results

### 5.1. Data Collection

In this section, we verify the validity of the proposed travel time PI construction method through experiments. The data set for this study is from the Shenyang Municipal Bureau of Big Data, while the ANN toolbox in MATLAB 2016a is introduced to train the neural networks. We choose the No. 239 bus line in Shenyang, China, as an example for experimental verification, as shown in Figure 4. The No. 239 bus line starts at Kang li Automobile Company in the west and ends at Quan yuan Community in the east. There are 26 stops on the entire journey with a total length of 13.9 km. Each vehicle is equipped with a GPS positioning device, which collects bus location data every 5 s and transmits it to the data center in real time. Each record includes line name, vehicle serial number, date of data generation, time of data generation, longitude, latitude, instantaneous speed, running direction, and the serial number of the next station. The serial number of the onboard machine on each bus is unique and corresponds to a fixed driver. This study only uses driving data in a single direction (that is, from west to east) to ensure the consistency of road conditions and other external conditions.

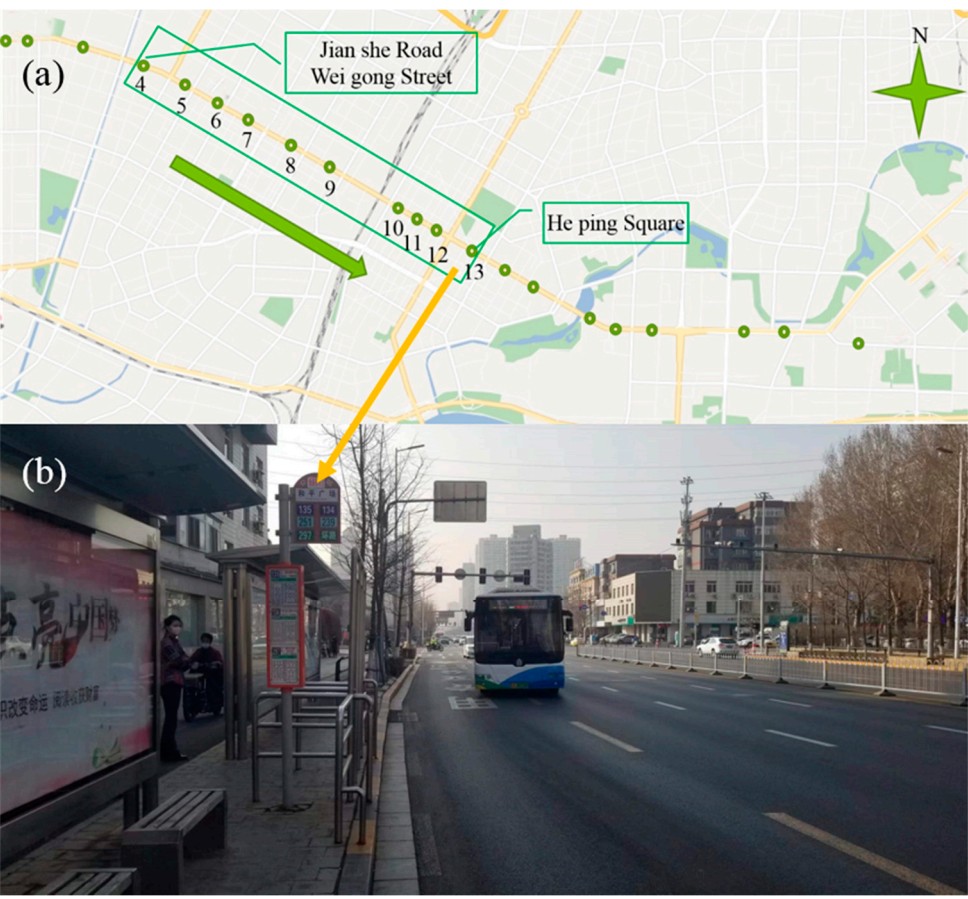

**Figure 4.** (**a**) Route map of bus No. 239, (**b**) photo of 239 Bus Route.

Due to the influence of GPS signal strength, the data from some sections of the road are seriously missing. Therefore, the road section from the fourth station (Jian she Road, Wei gong Street) to the 13th station (Peace Square) was intercepted as the research object, which is composed of nine adjacent stations. The predicted road section is marked in Figure 4. For example, road Section 5 represents the road section between the 4th station (Jian she Road, Wei gong Street) and the 5th station (Jian she Road, Bao gong Street), and road Section 6 represents from the 5th station (Jian she Road, Bao gong Street) to the 6th station (Jian she Road, Xing shun Street), etc.

### 5.2. Driving Style Clustering of Bus Drivers

In this section, we select the travel time data from 9 drivers of bus No. 239 as the research object, and cluster the travel time data of 9 drivers according to the time division scheme proposed in Section 4.2. The driver number and the corresponding clustering sample number are shown in Table 1.

**Table 1.** Driver's number and corresponding clustering sample number.

| Driver's Number | 902334 | 902335 | 902340 | 902347 | 902349 | 902351 | 902353 | 902355 | 902359 |
|---|---|---|---|---|---|---|---|---|---|
| Clustering sample number | 1 | 2 | 3 | 4 | 5 | 6 | 7 | 8 | 9 |

The clustering results are shown in Figure 5. It can be seen that in the morning peak period from 7:00 to 9:00, the driving styles of drivers numbered 902334, 902355, 902349, and 902335 are similar, and the driving styles of drivers numbered 902353 and 902359

are similar, while the driving styles of drivers numbered 902347, 902351, and 902340 are independent, and are quite different from those of other drivers. In the other time periods, the clustering results are relatively similar. The driving styles of drivers numbered 902335, 902353, and 902359 are similar, the driving styles of drivers numbered 902334, 902349, and 902355 are similar, and the driving styles of three drivers numbered 902347, 902351, and 902340 are also independent, and are quite different from those of other drivers.

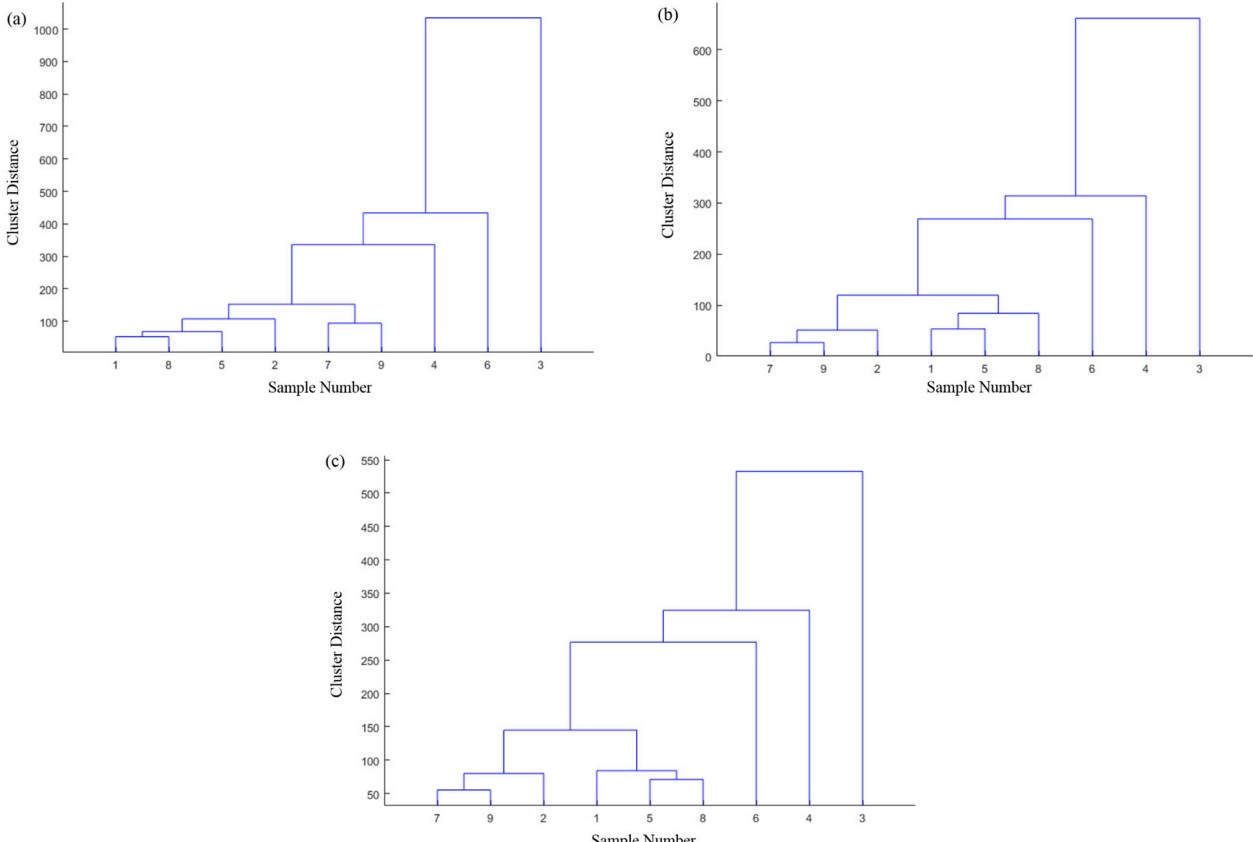

**Figure 5.** Clustering results for three time periods, (**a**) 7:00–9:00 (peak time), (**b**) 9:00–16:00 (off−peak time), (**c**) 16:00–19:00 (peak time).

### 5.3. PI Assessment Indexes

The performance of PIs is usually evaluated by two different measures, namely, coverage probability and PI width. PI coverage probability (*PICP*) refers to the ability of the constructed PIs to capture the actual target variables [3]. *PICP* can be expressed mathematically as:

$$PICP = \frac{1}{N} \sum_{i=1}^{N} C_i \tag{11}$$

where:

$$C_i = \begin{cases} 1, & if \ t_i \in [L_i, \ U_i] \\ 0, & if \ t_i \notin [L_i, \ U_i] \end{cases} \tag{12}$$

where $N$ is the number of samples in the test data set, $t_i$ represents the actual goal, $L_i$ and $U_i$ are the upper and lower limits of the ith PIs, respectively. The range of *PICP* is from 0% (when no target is surrounded by PIs) to 100% (when all targets are surrounded by PIs).

The important point is that *PICP* itself cannot describe all the characteristics of PIs. In fact, this only indicates the coverage of PIs to the target. If the PI is selected as the extreme value of the target, perfect *PICP* (100% coverage) can always be achieved. In fact, too−wide PIs are useless because they do not carry information about target changes. Therefore, an index must be defined to quantify the width of PIs. The average PI width (*MPIW*) is defined as follows [3]:

$$MPIW = \frac{1}{n}\sum_{i=1}^{n}(U(X_i) - L(X_i)) \tag{13}$$

Assuming that the target range *R* is known (*R* represents the range of maximum and minimum values obtained by the target during the whole prediction period), the normalized *NMPIW* [3,47] can be calculated as follows:

$$NMPIW = \frac{MPIL}{R} \tag{14}$$

Standardization of target ranges allows objective comparisons of PIs for different targets. In fact, *NMPIW* is a dimensionless measure that expresses the average width of PIs as a percentage of the base target range. When the extreme target value is used as the upper and lower limits of PIs, both *NMPIW* and *PICP* will be 100%, which indicates that there is a direct relationship between *PICP* and *NMPIW*. Under the same conditions, a higher *NMPIW* usually leads to higher *PICP*.

Ideally, *PICP* is at least equal to or higher than its nominal value and *NMPIW* is as small as possible (narrow PIs). Theoretically, these two goals are conflicting. A combination measure is needed to carry quantitative information about the scope of PIs and the extent to which they cover the target. Therefore, a coverage width criterion (*CWC*) composed of *PICP* and *NMPIW* was developed [47]:

$$CWC = NMPIW * (1 + \gamma(PICP)e^{(-\eta(PICP-\mu))}) \tag{15}$$

where $\gamma(PICP)$ is given by:

$$\gamma = \begin{cases} 0, & PICP \geq \mu \\ 1, & PICP < \mu \end{cases} \tag{16}$$

$\eta$ and $\mu$ in (15) are two hyperparameters controlling the location and amount of the *CWC* jump. Note that *CWC* is a negatively oriented unique skill score; the smaller, the better [22].

*5.4. Results and Discussions*

Based on the experimental comparison of different levels, the clustering results of the fourth level are selected from top to bottom. In addition, based on the clustering results of Section 5.2, drivers numbered 902335, 902353, and 902359 are selected as research objects. Their corresponding travel time data comes from the driving data of working days from 4 January 2016 to 22 January 2016. The preprocessed data are divided into three sample sets, D1, D2, and D3, of which D1 and D2 are used to train the travel time prediction model, and D3 is the test sample. The data of 10 days from 4 January 2016 to 15 January 2016 are used as the training data set, the data from 18 January 2016 to 20 January 2016 are used as the sample set, and the data from 21 January 2016 to 22 January 2016 are used as the D3 test set.

The departure schedules of the three drivers to be predicted in the D3 test set from 21 January 2016 to 22 January 2016 are shown in Table 2.

**Table 2.** Departure time of 3 drivers predicted in data set D3.

| Time Period | 20160121 | | | 20160122 | | |
|---|---|---|---|---|---|---|
| | 902335 | 902353 | 902359 | 902335 | 902353 | 902359 |
| 7–9 | 7:23:07 | 8:33:09 | 7:13:38 | 8:58:02 | 8:20:33 | 7:05:45 |
| 9–12 | 10:06:04 | | 9:57:11 | 11:55:28 | | 9:37:12 |
| 12–16 | 14:19:48 | 12:00:41 15:37:40 | 12:57:05 15:14:05 | 14:38:04 | 15:51:56 | 12:45:11 |
| 16–19 | 16:42:22 | 18:11:16 | 17:36:00 | 17:00:06 | 18:38:56 | 15:58:15 |

Their travel time PIs were constructed before and after clustering. Hereafter, for the sake of simplicity, P and HC subscripts are used to represent personalized and hierarchical clustering PI results, respectively. For example, $PI_P$ for 902335 represents the personalized PI result of the driver numbered 902335. $PI_{HC}$ for 902335 represents the PI result of driver number 902335 after hierarchical clustering.

The MATLAB software package was used for PI construction to avoid incorrect results caused by random initialization of NN parameters. Each group of experiments was repeated 20 times and then the average value was calculated as the final result. Two experiments on confidence levels (80% and 90%) were carried out to compare the effect of PIs. The results are shown in Figures 6 and 7, respectively.

When the confidence level was set to 80%, the results were as follows (see Figure 6).

It is not difficult to see from Figure 6 that when the confidence level is set to 80%, the results of $PI_{HC}$ and $PI_P$ can include most targets. At the same time, it can also be seen from Figure 6 that the width of $PI_{HC}$ is significantly narrower than that of $PI_P$. This shows that the effect of the PI model after clustering is better. In addition, few targets of the prediction results of these two models fall outside the interval. We also found that most of the targets that fall outside the PIs are located in the morning peak (7:00−9:00) and evening peak (16:00–19:00) periods, which have the characteristics of the impact of emergencies.

The precision of the prediction intervals results can be accurately compared by calculating the interval coverage probability *PICP*, interval average width *MPIW*, standardized interval average width proportion *NMPIW*, and comprehensive coverage width standard *CWC*.

Table 3 summarizes the *PICP*, *MPIW*, *NMPIW*, and *CWC* for the PIs, when the confidence level was set to 80%. According to the results in Table 3, the *PICP* of each interval exceeds the confidence level set (80%). The *PICP* value of $PI_{HC}$ was greater than or equal to that of $PI_P$ in all cases. The *MPIW* value of $PI_{HC}$ is smaller than the *MPIW* value of the corresponding $PI_P$ (23.33, 54.24, and 28.61 in three cases, respectively), which indicates that the PIs after clustering are narrower without reducing the coverage probability (*PICP*). The corresponding *NMPIW* values also decreased (decreased by 18.93%, 10.39%, and 14.19%, respectively). At the same time, the *CWC* value of $PI_{HC}$ in each case was also smaller (decreased by 18.93, 10.39, and 14.19, respectively). This further shows that the PI quality after clustering is better.

When the confidence level was set to 90%, the results are shown in Figure 7. With the increase in confidence level, the interval width of $PI_P$ and $PI_{HC}$ becomes wider than when the confidence level is 80%. However, the width of PI results after clustering is also narrower than that predicted by single−driver data.

Table 4 summarizes the *PICP*, *MPIW*, *PINAW*, and *CWC* for the PIs, when the confidence level was set to 90%. From Table 4, it can be seen that with the increase om confidence level, the values of *PICP*, *MPIW*, *PINAW*, and *CWC* of $PI_P$ and $PI_{HC}$ all have different degrees of increase than when the confidence level is 80%. Likewise, the quality of the PIs after clustering at the 90% confidence level is significantly better than the quality of the PIs from the data of a single driver.

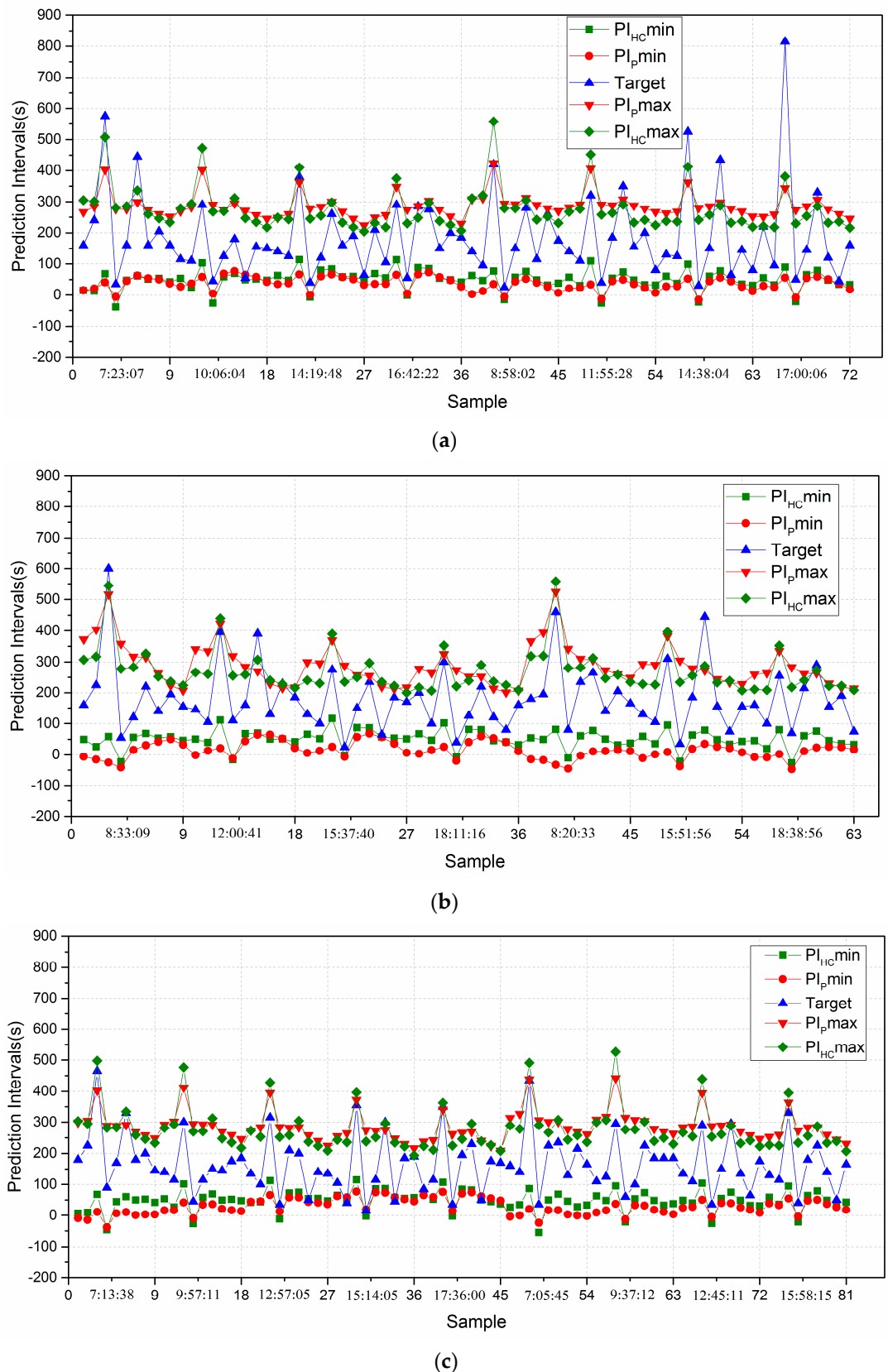

**Figure 6.** PI results at 80% confidence level: (**a**) $PI_{HC}$ and $PI_P$ for 902335; (**b**) $PI_{HC}$ and $PI_P$ for 902353; (**c**) $PI_{HC}$ and $PI_P$ for 902359.

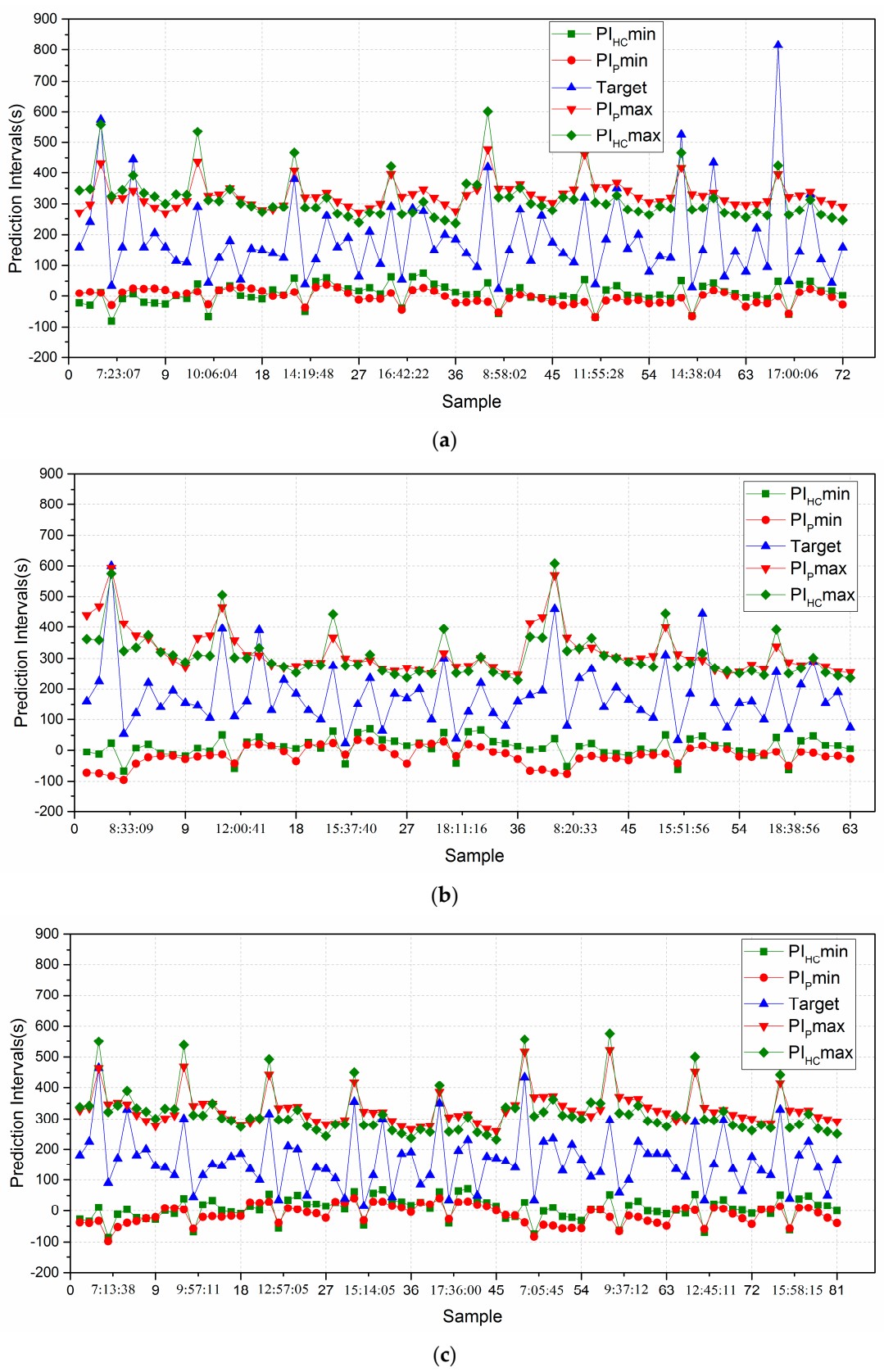

**Figure 7.** PI results at 90% confidence level: (**a**) PI$_{HC}$ and PI$_{P}$ for 902335; (**b**) PI$_{HC}$ and PI$_{P}$ for 902353; (**c**) PI$_{HC}$ and PI$_{P}$ for 902359.

**Table 3.** PIs characteristics for test samples when the confidence level is 80%.

| Case Study | Model | PICP (%) | MPIW | NMPIW (%) | CWC |
|---|---|---|---|---|---|
| 902335 | PI$_P$ | 87.50 | 250.69 | 57.05 | 57.05 |
| | PI$_{HC}$ | 87.50 | 227.36 | 38.12 | 38.12 |
| 902353 | PI$_P$ | 92.06 | 274.30 | 48.05 | 48.05 |
| | PI$_{HC}$ | 92.06 | 220.06 | 37.66 | 37.66 |
| 902359 | PI$_P$ | 88.89 | 257.77 | 53.66 | 53.66 |
| | PI$_{HC}$ | 93.83 | 229.16 | 39.47 | 39.47 |

PI$_P$—Personalized PIs, PI$_{HC}$—Hierarchical Clustering PIs.

**Table 4.** PIs characteristics for test samples when the confidence level is 90%.

| Case Study | Model | PICP (%) | MPIW | NMPIW (%) | CWC |
|---|---|---|---|---|---|
| 902335 | PI$_P$ | 93.06 | 332.05 | 60.79 | 60.79 |
| | PI$_{HC}$ | 88.89 | 310.48 | 45.51 | 115.90 |
| 902353 | PI$_P$ | 95.24 | 336.79 | 48.83 | 48.83 |
| | PI$_{HC}$ | 95.24 | 298.43 | 44.23 | 44.23 |
| 902359 | PI$_P$ | 98.77 | 343.49 | 55.49 | 55.49 |
| | PI$_{HC}$ | 100.00 | 314.49 | 47.49 | 47.49 |

PI$_P$—Personalized PIs, PI$_{HC}$—Hierarchical Clustering PIs.

From the results of Tables 3 and 4, we found that under the same confidence level, there are significant differences in the prediction results under different driving style groups. For example, the quality of the predicted PIs for the driver numbered 902353 is better than that of the PIs numbered 902335 and 902359. The reason for this difference may be due to different driving styles of drivers. For example, drivers with mild driving styles have relatively stable daily travel times without significant fluctuations, As a result, the prediction interval is narrower and the prediction effect is better. Drivers with more aggressive driving styles have a large fluctuation in their daily travel time, resulting in a wide prediction range and showing poor prediction results.

At the same time, we also found from Figures 6 and 7 that during peak hours, some target values exceed the predicted range, resulting in a decline to some extent in the accuracy of the model. This indicates that bus travel times can be affected by other reasons during peak hours, such as traffic accidents and private cars or bicycles occupying the bus lanes, which indicates that our model still lacks the ability to respond to emergencies. Our next study will focus on analyzing the impact of unexpected events on the accuracy of the PIs, further optimizing our model and improving the accuracy of the prediction. This will help to make public transportation more popular with travelers to reduce the number of private cars and address the ecological and environmental problems caused by the increase in the number of private cars [48].

## 6. Conclusions

The purpose of this paper is to quantify the uncertainty associated with the prediction of the bus travel time point. We proposed a personalized PI model for bus travel time based on the bootstrap method, which is used to construct personalized PIs for driver travel times with different driving styles. It is vital to help passengers decide departure times at bus stops and thereby reduce the anxiety of waiting passengers. To further improve the quality of PIs, we optimized the training data set by considering the driver's driving style factors, and then constructed a bus travel time PI model based on driver driving style clustering and the bootstrap method. We carried out experimental verification on a real bus line data set in Shenyang, China, and evaluated the performance of the proposed prediction model. Two different confidence intervals of 80% and 90% were used to evaluate the coverage characteristics of the two models. At the same time, through the quantitative evaluation (*PICP*, *MPIW*, *NMPIW*, and *CWC*) of the constructed PIs, the *PICP* of each interval exceeded the confidence level set (80%). It was also found that the quality of the

prediction intervals constructed by clustering the driving style data is better (the *MPIW* values decreased by 23.33%, 54.24%, and 28.61 respectively, and the corresponding *NMPIW* values also decreased by 18.93%, 10.39%, and 14.19%, respectively), and the *CWC* value of $PI_{HC}$ in each case was also smaller (decreasing by 18.93, 10.39, and 14.19, respectively). The effects of the two PI models were further analyzed.

Our future research will consider how to continue to improve the quality of the PIs of the bus travel time. We will consider more variables that affect travel time in the prediction model, such as traffic accidents, number of passengers, extreme weather, and other factors. Our research can also be applied to personalized travel time interval prediction for taxis or private cars when relevant data sets are available. Research on bus travel time interval prediction has received less attention from scientists, and new technologies need to be developed in the future to further improve the quality of bus interval prediction, with the aim of improving the attractiveness of buses.

**Author Contributions:** Z.Y. and B.Z. contributed to the study conception, the design of the experiments, and the paper's structure. Z.Y. performed experiment analysis and wrote the first draft of the manuscript. Z.Y. and B.Z. participated in the revision and proofreading of the paper. All authors have read and agreed to the published version of the manuscript.

**Funding:** This work was supported by the Key Project of National Natural Science Foundation of China (U1908212), Central government guided local science and Technology Development Fund Project (1653137155953), and Liaoning Province "takes the lead" science and technology research project (2021jh1/10400006).

**Data Availability Statement:** The data presented in this study are available on request from the corresponding author.

**Conflicts of Interest:** The authors declare no conflict of interest.

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
