# Peer review of "Construction of Personalized Bus Travel Time Prediction Intervals Based on Hierarchical Clustering and the Bootstrap Method"

_electronics, doi:10.3390/electronics12081917_

Round 1
Reviewer 1 Report
The manuscript tried to predict the bus travel time by using hierarchical clustering and the bootstrap method. It is an interesting work that may attribute to city construction. However, some points need to be improved.
Abstract: in Lines 23-24, the authors stated that "the two models constructed in this paper can effectively quantify the uncertainty of the point prediction results.", can you state by specific numerical data how accurate it is?
Line 44-43: this sentence did not make sense, there is no verb. Please rewrite it.
Sections 2 and 3 should be in the introduction, to review the previous similar work. Please integrate it with an introduction but be concise. It is recommended to be within 2 pages for the Introduction.
The material and methods section (which looks like section 4) is not clearly described. Where the data used for the model come from should be described. Do you use any software to run your model? If so, please state it in the material and methods section.
Table 1: it is very hard to understand the meaning of code and numbers. As previous numbers are shown in Figure 4, is there any relationship between the two?
Why are there no legend labels for x and y-axes? the meaning of x- and y-axes are not clear.
How the accuracies of your models are not clear. How do you validate them?
Table 4: the abbreviations should be explained as a note.
Reviewer 2 Report
There is not much to tell on this interesting manuscript, but some improvements would definitely increase its attractiveness among the readers. These are:
1) to better understand the driving styles on route 239, some photos showing the route would help assess whether the quality of the streetscape/urban environments might affect drivers' behaviors. Ditto for seasonal effects. Please, elaborate environmental/behavioral conditions. Some details on the types of vehicles would help, too
2) r. 160-167 - here is not clear the reasons why the authors prefer the bootstrap method over the others there mentioned; major clarifications are required
3) Discussion is missing, as results are just commented. For example, stress the potential or the environmental consequences of ecodriving. Several projects funded by the European Commission tested ecodriving, consider results reported in e.g. Corazza M.V., Guida U., Musso A., Tozzi M. From EBSF to EBSF-2: A compelling agenda for the bus of the future: A decade of research for more attractive and sustainable buses. (2016) EEEIC 2016 - International Conference on Environment and Electrical Engineering, art. no. 7555479. DOI: 10.1109/EEEIC.2016.7555479.
More can be added about the transferability of these results elsewhere, etc.
4) Conclusions are a proxy for a short summary, whereas they should include remarks on this study's caveats (if any), future developments, methodological challenges, etc.
Reviewer 3 Report
My comments are:
1. Please provide some quantified results in the abstract. "Effectively quantify" - this is too vague. The same for the conclusions. Use Table 3 for values.
2. Figure 5 needs a more descriptive caption and the axes need labels.
3. Overall, I think the paper is ready for acceptance. I would like to see a more detailed discussion of the results in 5.4. There is very little text and a large number of figures with many interesting features and trends. Why not discuss some of these? Any outliers? Was anything unexpected? And so on.
Round 2
Reviewer 1 Report
The authors have addressed most of my comments. Conclusions should be the main finding of the manuscript, it is not the discussion. Therefore, the references should not appear in the conclusions (in Line 538)
Author Response
Response to Reviewer 1 Comments
Point 1: The authors have addressed most of my comments. Conclusions should be the main finding of the manuscript, it is not the discussion. Therefore, the references should not appear in the conclusions (in Line 538).
Response 1:
Thanks a lot for Reviewer’s comments. We have made revisions to the conclusion according to the suggestions of the reviewer and placed reference [47] in the 5.4 Results and Discussions section. Please see “Conclusions”, page 17 of the new manuscript (marked version-R2).

Reviewer 2 Report
The Authors met all the revision requirements and now the manuscript is it for publication. Congrats to the Authors
Author Response
Thank you very much for the reviewer's comments.